# Development of a Device for Defatting Full Skin Grafts Through Mechanical Defatting in Children and Adolescents

**DOI:** 10.3390/ebj6030044

**Published:** 2025-08-14

**Authors:** Philipp Christoph Köhler, Helen Glosse, Steffan Loff, Raphael Staubach

**Affiliations:** Department for Pediatric Surgery, Klinikum Stuttgart/Olgahospital, Kriegsbergstraße 60/62, 70174 Stuttgart, Germany

**Keywords:** full-thickness skin grafts, mechanical defatting, reconstructive surgery

## Abstract

Full-thickness skin grafts are a cornerstone in reconstructive surgery for extensive skin defects, particularly in pediatric patients, where rapid vascularization is essential for successful engraftment. Traditional defatting methods using scalpels and scissors are labor-intensive and increase the risk of graft or operator injury. To improve efficiency and safety, a mechanical defatting device called LOMA (named after the inventors Loff and Maja) was developed at Klinikum Stuttgart. This study evaluates the first 28 transplants performed with it, assessing graft outcomes using the POSAS and comparing physical properties of the grafts with those of healthy contralateral skin, ankle skin, and palmar skin using DermaLab Combo’s ultrasound and elasticity probes. Results showed that grafts prepared with LOMA exhibited similar physical characteristics to contralateral healthy skin. Differences in elasticity were observed when compared to ankle skin, and significant disparities were found when compared to palmar skin. POSAS scores averaged 3.3 from patients and 2.2 from physicians, indicating satisfaction with functional and aesthetic outcomes. The findings support the effectiveness of full-thickness skin grafts, particularly when prepared using the LOMA system. Further multicenter studies are recommended to compare LOMA-prepared grafts with those using conventional techniques to quantify the added value of this mechanical defatting approach.

## 1. Introduction

FTSGs are an essential method in reconstructive surgery for treating extensive skin defects or performing corrections when conservative treatment has failed. Damaged skin can be replaced by grafting healthy skin from another part of the body [1], a process particularly crucial in children since they grow and have a higher risk of developing skin contractures [1,2].

Full-thickness skin transplantation is commonly used for correcting post-burn scars, after skin tumor excisions, and for chronic pressure sores. However, it is less suitable for acute skin lesions, as healing times are longer and the risk of infection is higher [2,3]. When selecting a donor site, factors such as graft size, color, and thickness are important considerations. Frequently chosen “hidden donor sites” include the abdominal fold, buttock fold, groin, and the mastoid area behind the ear [2,4].

The healing process of FTSGs relies on the rapid vascularization of the donor skin from the underlying subcutaneous tissue. Inosculation, the critical phase when small blood vessels from the graft connect and merge with the recipient site’s vessels, is essential for successful transplantation [2] (Figure 1). However, shear forces can hinder this process, with a deleterious impact on graft survival. To achieve proper graft healing, the wound bed must be free of infection and any avital tissue [2,3]. To enable the FTSG to grow in, defatting of the graft is crucial. Since, on the other hand, excessive thinning of the FTSG can cause significant shrinkage of the transplant, precise removal of subcutaneous fat from the FTSG is of utmost importance [4].

Currently, skin is prepared for grafting with scalpels and scissors to thin and defat the skin while it is stretched over a finger. This technique poses the risk of damaging the graft and failing to achieve the desired thickness. In addition, this process is time-consuming, even for experienced surgeons, and also poses a risk of injury to the surgeon [1].

FTSGs are typically secured with pressure bandages, essential for exerting adequate pressure and minimizing shear forces. Proper fixation ensures optimal graft adherence, facilitating direct contact with the wound bed.

To enable the rapid and reliable production of clinically flawless FTSGs, a device for the defatting of FTSGs has been developed over several years at the Department of Pediatric Surgery at Klinikum Stuttgart.

### 1.1. Development

At the Department of Pediatric Surgery at Klinikum Stuttgart, full skin graft transplantation is a well-established technique, especially for treating skin contractures in young children due to burn wounds. The defatting process is the most time-consuming part of the procedure, posing risks of graft perforation and injury to the surgeon. To address these challenges, a mechanical defatting machine was developed to expedite the process, reduce operation times, and minimize risks.

The initial prototype of the skin transplant defatting machine was adapted from a professional meat skinning machine, featuring a hand crank for precise control. Initial tests using piglet skin highlighted issues with the crimper, which caused superficial cuts and required excessive force. To resolve these issues, a specialized crimper with custom-rounded edges was developed, and a gearbox was integrated to significantly reduce the required force (Figure 2). Following these modifications, we obtained intact skin grafts. Subsequent testing with human skin remnants confirmed the effectiveness of the modified machine, called LOMA (named after the inventors Loff and Maja) skin defatting machine.

### 1.2. Functional Mechanism of the LOMA

The crimper secures the epidermis and dermis while the skin is manually guided over a rotating roller toward a fixed blade that removes the underlying fat tissue (Figure 3). The pulling direction is adjusted to ensure a uniform cutting plane. Thanks to an integrated gearbox, the device can be operated manually with significantly reduced force, allowing for precise and controlled handling of the graft. At the end of the process, a fully defatted transplant with a smooth cut surface and a separate flap of excess subcutaneous fat is obtained. Depending on clinical needs, the graft thickness can be adjusted starting from a minimum of 4 mm. The LOMA is a certified medical device with the CE mark, meeting European safety and quality standards. Additionally, it is also protected by patents in Germany and throughout Europe, securing its innovative design and technology.

### 1.3. Skin Analysis

#### 1.3.1. Skin Examination and Assessment

Research on the mechanical properties of skin is hindered by a lack of standardized experimental protocols. The variety of methodologies used complicates direct comparisons and hampers accurate interpretation of findings. Therefore, establishing standardized procedures and enhancing methodological consistency are crucial for gaining clearer and more reliable insights [5,6].

The DermaLab Combo, developed by Cortex Technologies in Denmark, addresses this need by providing standardized skin measurement tools. Though primarily designed for cosmetic applications, it is widely used in medical research [7].

The latest DermaLab Combo includes up to 11 probes for various skin analyses, such as high-resolution ultrasound, elasticity, hydration, and pH assessment. In this study, the focus was on ultrasound and elasticity probes to assess the patients’ skin status [7,8].

#### 1.3.2. Ultrasound Probe

The 20-megahertz ultrasound probe provides detailed imaging of the skin’s layers, allowing for precise assessment of skin thickness and structure (Figure 4). By emitting high-frequency sound waves, it creates real-time, high-resolution images to visualize various skin components. The intensity of the received signal is represented by color: dark colors indicate areas with minimal reflection and homogeneous composition, while lighter colors indicate strong reflections and heterogeneous composition [7].

The ultrasound image displays the epidermis on the left, the subcutis on the right, and, in between, the dermis. The latter shows varying intensities, reflected in different colors, while the subcutis exhibits low-intensity areas due to its homogeneous fatty composition [7].

The ultrasound image provides “curve” measurements, with red indicating intensity through the water chamber and subcutaneous layer, yellow representing the low echogenic band of the dermis, and green showing the more reflective deeper dermis. These curves combined indicate average skin thickness, while diseased skin displays different patterns, with uniform areas appearing dark and non-uniform structures showing varying reflections [7].

#### 1.3.3. Skin Elasticity Analysis

The elasticity measurement is based on suction applied to the skin surface. Negative pressure of 15,000/40,000/65,000 pascal, reflecting the soft/normal/firm setting, can be applied to the analyzed skin area. For skin elasticity measurements, three repetitive cycles are performed. Three cycles are used in order to obtain a validity check for proper probe-skin application. Repeated measurement, however, inevitably leads to a gradual increase in elevation and retraction time with each cycle, resulting in a slight decrease in E and VE after each iteration [7].

Young, smooth skin elevates and promptly retracts upon suction and release. Conversely, aged, fibrotic, and scarred skin elevates with ease but demonstrates a slower retraction response. Therefore, what is usually considered to be skin elasticity is a more complex feature and can be quantified by analyzing the elevation and the retraction phase [7] (Figure 5).

The SkinLab Combo offers three descriptive parameters for skin elasticity:Young’s elasticity modulus (E);Skin retraction time (R);Viscoelasticity (VE).

##### Young’s Elasticity Modulus (E)

E is a pivotal mechanical attribute characterizing the rigidity of a material when subjected to tensile or compressive forces. It quantifies the relationship between stress (σ), denoting force per unit area, and strain (ε), representing proportional deformation, within the range of linear elasticity [10] (Figure 6).(1)$$E=\frac{\sigma }{\varepsilon }$$

##### Skin Retraction Time (R)

R is defined as the duration in milliseconds for the skin to retract from its peak elevation to 33% of that peak elevation. This metric is displayed in the red segment of the elasticity screen graph [7].

##### Visco-Elasticity (VE)

Dividing E by the retraction time R yields a parameter called VE, which takes into account both the elevation and retraction phases. R is normalized by a retraction time of 260 ms (representing normal VE in 28–60-year-olds) [7].(2)$$VE=\frac{E}{R}$$

##### Elasticity Measurement

Each of the patient’s individual elasticity measurement cycle can be further divided into subsegments, the so-called *U*-values of the suction/retraction curve (Figure 3). The following subdivisions are chosen:(Ue)—immediate (0.1 s) deformation or skin extensibility (mm)(Uv)—delayed distension (mm)(Uf)—final deformation (mm)(Ur)—immediate (0.1 s) retraction (mm)(Ua)—total recovery (mm)Point 0—startPoint a—time at maximum elongationPoint b—final elongation after returning to normal pressure

The *U*-values provide detailed information about every phase of the elongation and retraction processes of the skin itself (Figure 7).

#### 1.3.4. POSAS

In addition to physical measurements, a POSAS was administered to patients or their guardians [11]. This two-part test included a patient self-assessment and an observer assessment by the attending physician. Patients evaluated parameters such as pain, itching, color, stiffness, thickness, irregularity, and their overall opinion of their full skin transplant, rating each on a scale from 1 to 10, where 1 represents the best outcome and 10 represents the worst [11]. The treating clinician completed the other part of the questionnaire, evaluating objective scar characteristics such as vascularity, pigmentation, thickness, relief, pliability, surface area, and overall appearance, using the same rating system from 1 to 10.

## 2. Methods

### 2.1. Patient Cohort

#### 2.1.1. Transplanted Patients

We retrospectively analyzed our digital patient register and identified 28 FTSG performed on 23 patients at the Klinikum Stuttgart.

#### 2.1.2. Healthy Comparisons

In addition, a control group was established comprising patients with healthy skin. These individuals were selected from inpatients of the Department of Pediatric Surgery at the Klinikum Stuttgart who had undergone procedures not affecting the skin. Inclusion criteria were age under 18 years, no known skin disease affecting the palms or legs, and written consent from their parents.

Measurements were taken both at the palm and the skin of the ankle. These two locations were chosen to represent different skin types. The palm typifies the tough, ridged skin found in areas subject to continuous mechanical stress and friction, while the skin of the ankle represents fine skin not exposed to these impacts. Furthermore, these locations were selected for their ease of access in fully clothed patients.

The skin measurements with the DermaLab Combo were taken on a voluntary basis, and the study was conducted following approval from the Ethics Committee II of the University of Heidelberg, ensuring adherence to ethical guidelines and standards in research.

### 2.2. Statistical Analysis

The aim of the statistical analysis was to demonstrate that LOMA-defatted FTSG does not differ significantly from healthy contralateral skin or corresponding areas in healthy controls.

We first assessed whether the differences in each paired sample followed a normal distribution using the Shapiro-Wilk test (α = 0.05).

Three comparisons were made:Transplant vs. contralateral healthy skin (paired)Transplant vs. healthy palm (unpaired)Transplant vs. healthy ankle (unpaired)

For normally distributed differences, we used paired *t*-tests or unpaired *t*-tests. If normality was not met, we applied Wilcoxon signed-rank or Mann-Whitney U tests. This ensured appropriate test selection based on data type and distribution.

The statistical analysis was conducted in collaboration with the medical statistics team at Heidelberg University. To process the data and generate results, SPSS v.Statistics 30, Microsoft Excel, and DATAtab were used as statistical and graphical software tools.

## 3. Results

Our patient cohort was identified through a retrospective review of the patient local database based on the corresponding surgical codes (5-925.xx). From May 2022 onwards, all full-thickness skin transplantations at the Department of Pediatric Surgery of the Klinikum Stuttgart were assisted by the LOMA skin defatting machine. A total of 35 transplantations were identified, performed in 23 patients. Among these patients, four declined participation in our study, two had skin transplants that were too small to measure (less than 1 × 1 cm^2^), and one skin transplant was in a spot where measurement was impossible (interdigital space). As a result, 28 FTSG from 17 patients were included in our analysis (Figure 8).

### 3.1. POSAS Score in the FTSG Group

#### 3.1.1. Patient Score

The POSAS score for patient assessment of FTSGs indicated relatively low levels of pain and itch post-transplantation, with average scores of 1.9 and 1.5, respectively, representing the best-rated aspects of the patients’ experience.

The skin color of the grafts was rated an average of 4.3. However, one patient who developed global hyperpigmentation of the FTSG rated it a 10, which significantly increased the overall average [12]. If this outlier was excluded, the average rating for skin color was 3.8. Nonetheless, it emerged as the category with the lowest average rating overall.

Other parameters, including skin stiffness, skin thickness, and skin irregularity, received average scores ranging from 2.8 to 2.9. Despite these variations, the overall patient opinion of the FTSG averaged 3.3 (Table 1).

#### 3.1.2. Observer Score

In addition to patient assessments, observer evaluations of the FTSG were conducted, focusing on various skin parameters. The results for observer-assessed skin vascularity were notably high in quality, with an average score of 1.1, indicating minimal visible blood vessels in the grafted skin. Skin pigmentation received a score of 2.1, suggesting slight color variation compared to the surrounding skin. Observer evaluations of skin thickness yielded a score of 2.2.

Skin relief, which refers to the texture and surface contour of the skin, was rated 2.0, showing good uniformity in surface appearance. Skin pliability, reflecting the skin’s flexibility and resilience, received a score of 2.5, indicating a moderate level of elasticity. The surface area of the grafted skin was evaluated at 1.9, suggesting only a slight discrepancy in size compared to the surrounding skin, which is close to ideal.

Overall, the observers’ general opinion of the FTSG was rated 2.2 (Table 2).

### 3.2. Analysis of Transplanted Skin Compared to Healthy Skin on the Patients’ Opposite Side

#### 3.2.1. Skin Thickness

The skin thickness of the FTSG group exhibited higher values in median (Mdn) with 1053 and mean (M) with 1110, compared to the healthy opposite side group (Mdn = 974.5, M = 975.9). The observed difference in the thickness of transplanted skin compared to healthy skin was not statistically significant, with a *p*-value of 0.097, indicating no substantial difference between the two.

#### 3.2.2. Viscoelasticity Analysis (VE)

The VE elasticity modulus of the transplant group had a mean (M) value of 9.4, while the healthy opposite side group had a higher mean value of 10. The median values were 6.4 and 6.7, respectively. The difference in VE between the two groups was not statistically significant (*p* = 0.07).

#### 3.2.3. Young’s Elasticity Modulus (E)

The Young’s modulus (E) of the FTSG group had a lower mean (M = 6.15) but a slightly higher median (Mdn = 4.95) compared to the healthy opposite side group (M = 7.48, Mdn = 4.85). A paired *t*-test revealed that this difference was not statistically significant (*p* = 0.28).

#### 3.2.4. Retraction Time

The retraction time of the transplant group (M = 186.1, Mdn = 169.5) was lower than in the healthy opposite side group (M = 191.8, Mdn = 171.5). The difference was not statistically significant (*p* = 0.461).

#### 3.2.5. *U*-Values

As with the other parameters, no statistically significant differences were detected for any of the U-values between the FTSG and the healthy opposite side groups (Table 3).

#### 3.2.6. Summary

In eight comparisons using the DermaLab Combo ultrasound and elasticity probes, all *p*-values exceeded 0.05, indicating no significant differences between the FTSG group and the healthy opposite side group (Figure 9).

### 3.3. Analysis of Transplanted Skin Compared to Palmar Skin in a Healthy Control Cohort

#### 3.3.1. Skin Thickness

The FTSG group had higher skin thickness values (M = 1110, Mdn = 1053) than the healthy palm group (M = 792, Mdn = 816). A significant difference between the two patient groups was found (*p* < 0.001).

#### 3.3.2. Viscoelasticity Analysis (VE)

The FTSG group had lower mean and median VE values (M = 9.4, Mdn = 6.4) compared with the healthy palm group (M = 13.6, Mdn = 11.5). This difference was statistically significant (*p* = 0.029).

#### 3.3.3. Young’s Elasticity Modulus (E)

The FTSG group had lower mean and median E values (M = 6.15, Mdn = 4.95) than the healthy palm group (M = 14.6, Mdn = 13.4), a statistically significant difference (*p* < 0.001).

#### 3.3.4. Retraction Time

The FTSG group had lower mean and median values for retraction time (M = 186.1, Mdn = 169.5) than the healthy palm group (M = 318, Mdn = 264). The *p*-value of 0.001 suggests a significant difference.

#### 3.3.5. *U*-Values

The figures for final skin deformation (Uf) and total skin recovery (Ua) showed no statistically significant differences between the FTSG and the healthy palm groups. All the other U-values in the analysis of FTSG and healthy palms were statistically significant (Table 4).

#### 3.3.6. Summary

In eight comparisons between the FTSG group and the healthy palm group, the *p*-values for Uf and Ua exceeded 0.05, indicating no significant differences. In contrast, the remaining six comparisons yielded *p*-values below 0.05, demonstrating significant differences between the FTSG group and the healthy palm group (Figure 10).

### 3.4. Analysis of Transplanted Skin Compared to Ankle Skin in a Healthy Control Cohort

#### 3.4.1. Skin Thickness

The FTSG group had a lower mean but higher median value (M = 1110, Mdn = 1053) than the healthy ankle group (M = 1156, Mdn = 1106). At 0.3305, the *p*-value was above the significance level, suggesting that the skin thickness was comparable between the two groups.

#### 3.4.2. Viscoelasticity Analysis (VE)

The transplanted group showed a higher mean but lower median VE value (M = 9.4, Mdn = 6.4) compared to the healthy ankle group (M = 8.62, Mdn = 7.45). The difference in VE was considered to be non-significant, with a *p*-value of 0.7044.

#### 3.4.3. Young’s Elasticity Modulus (E)

Concerning E, the FTSG group showed higher mean and median values (M = 6.15, Mdn = 4.95) compared to the healthy ankle group (M = 4.35, Mdn = 3.75). However, the *p*-value of 0.1029 indicated that this difference was not statistically significant.

#### 3.4.4. Retraction Time (RT)

In the transplanted group, higher values for RT were found (M = 186.1, Mdn = 169.5) compared to the healthy ankle group (M = 128, Mdn = 126); the *p*-value of 0.0004 suggested a significant difference between the groups.

#### 3.4.5. *U*-Values

The analysis of the U-values revealed significant differences between the transplant group and the healthy ankle group across all U-value skin parameters (Table 5).

#### 3.4.6. Summary

In the analysis of skin thickness, VE, and E, the *p*-values exceeded the significance level of 0.05, indicating no significant differences. However, in the case of skin retraction time and U-values, the *p*-values were below the significance level, i.e., the differences between the two cohorts were statistically significant (Figure 11).

## 4. Discussion

The goal of our study was to comprehensively evaluate the quality of intraoperatively prepared full-thickness skin grafts (FTSG) using the LOMA skin defatting machine. Our assessment combined objective measurement parameters obtained with the DermaLab Combo Ultrasound and Elasticity probe and subjective feedback from both patients and examiners by means of a detailed questionnaire. Integrating both objective and subjective assessments, we aimed to provide robust data on the outcome of full-thickness skin transplantation using this machine-based preparation technique.

Full-skin transplantation competes with various other covering techniques, such as artificial dermal skin substitutes in combination with split skin transplantation which have gained prominence in recent years [13,14,15,16,17]. Despite the availability of alternative methods, specific scenarios necessitate full skin transplantation, particularly in sensitive areas such as the palms or regions under significant strain, such as prosthesis stumps [15,16,18,19]. Our study data support this approach, with 18 FTSG procedures performed on fingers or palms and six on patients’ lower extremities as recipient sites.

Overall patient satisfaction with the transplants was high. A total of 71% of our transplanted patients rated their experience with a score of three out of ten, with a score of one representing completely healthy skin. Similarly, the treating physicians’ evaluation of all FTSGs reached a score of at least three and, in a quarter of the cases, an ideal score of one.

The statistical analysis confirms the satisfactory POSAS scores reported by patients, their families, and the observing physicians. No statistically significant differences were found between the FTSG and the healthy opposite side groups across all measured parameters, with *p*-values exceeding 0.05 in all eight comparisons. These findings suggest that there is no difference between the FTSG and the healthy skin on the opposite side in terms of the measured parameters, including skin thickness and elasticity. In other words, skin transplanted utilizing the LOMA machine proved to show good functionality.

Comparative analysis between FTSG and healthy palmar skin revealed significant differences in seven of nine parameters, particularly those related to skin thickness and elasticity. The seven parameters showing significant differences emphasize structural and biomechanical discrepancies between the groups. No differences were found in final skin deformation (Uf) and total skin recovery (Ua). These differences are probably due to the finer nature of abdominal or gluteal donor skin compared to the mechanically robust ridged skin at the recipient site.

Furthermore, comparisons between FTSG and healthy ankle skin in a control cohort revealed notable differences in skin elasticity but not in skin thickness. This may be explained by the fact that all FTSGs were consistently prepared with a total skin thickness of 4 mm using the LOMA machine, while skin elasticity is influenced by multiple factors and cannot be affected by the defatting process of the LOMA machine itself. Of note, the comparison of FTSG skin with healthy palms and healthy ankles showed differences in several parameters that were not seen in the comparison between transplanted skin and that on the opposite side of the body. It is presumed that this is due to the differences in the skin properties in different regions of the body [20,21].

Analyzing our data, we also tried to ascertain whether using LOMA-defatted FTSG help to restore the functionality and esthetic appearance of compromised skin areas. Although the application of FTSG is a well-known method, the preparation technique reported here is new. Even though our results indicate very high standards of our FTSG, further research comparing FTSG prepared and defatted with the commonly applied hand defatting method with those defatted using the LOMA machine is necessary to provide further valuable information on the effect of the LOMA. The data comparing the transplanted with the healthy opposite site and the respective patient questionnaire are encouraging, as they show that functionality and appearance of the transplanted areas resemble that of healthy skin.

Apart from the positive effect on the quality of the transplanted skin, the LOMA technique offers several advantages in daily clinical practice. First, the machine rapidly removes the fatty tissue from the skin, so the defatting process takes just a few seconds. Second, the free FTSG can be implanted immediately after explantation from the donor site, saving valuable operating time, surgical supplies, and effort by the medical personnel. Specifically, it reduces the duration of the surgery, minimizes the use of additional instruments and consumables, decreases the workload and time required by surgeons, nurses, and other medical staff, reduces the amount of anesthesia needed due to the shorter operation time, and thus lowers the overall utilization of hospital facilities and resources. Our local data revealed that the average operation time was reduced by 69 min, from 174 min to 105 min when using the LOMA compared to defatting by hand. The sterile setup of the LOMA must be completed before use, which is ensured by the staff of our department’s operating room right before the operation starts. The LOMA machine eliminates the need for dividing work in terms of working simultaneously on the defatting process and the wound, saving valuable and costly resources, for instance a second physician working in parallel and, as mentioned, a longer operating time. While the acquisition of the LOMA requires an economic investment first, these costs are offset in long term by savings in anesthesia and personnel costs if use of the LOMA machine becomes the standard FTSG procedure.

An additional advantage of the LOMA over the commonly used manual defatting process is its precision in preparing FTSGs. Applied properly, the LOMA ensures high accuracy in FTSG surface preparation, minimizing the risk of partial damage of the graft caused by imprecise scissor strokes. Furthermore, a graft evenly prepared across the complete area facilitates inosculation, the process where blood vessels from the transplanted skin and the recipient site connect and fuse. This connection is crucial for establishing blood flow to the graft, which is necessary for its survival, ingrowth and functionality. A minor drawback of the LOMA technique is the incomplete dissection of the subcutaneous fat on the borders of the FTSG. During the dissection process, a small rim of subcutaneous fat can escape on both sides of the turning crimper resulting in a narrow fatty column of approximately 1 mm in width. This can easily be clipped off with a single scissor stroke once the defatting process in the LOMA machine is completed. Finally, since the FTSG defatting machine is constructed from solid steel, it is reusable after sterilization and extremely durable, but it is relatively heavy at 15.1 kg.

While the LOMA machine offers a significant advantage in facilitating the process of full skin transplantation, accurate evaluation of the indication for full skin transplantation is crucial. Alternative treatment options, such as dermal regeneration templates in combination with split skin grafting, are increasingly used, especially in patients with large-scale burns. While FTSG provides excellent skin quality, its usage is constrained by its expandability and donor site choices. For aesthetic reasons suitable donor sites must be selected carefully, such as the groin area beneath the waistband of trousers, the gluteal fold or the mastoid skin behind the ears. In our department, the majority of transplants is taken from the groin, since skin quality there is frequently optimal with a thick subcutaneous fat layer, that can easily be disassembled from the underlying fascia [2].

A drawback of FTSG from the groin area is the slight tendency for hyperpigmentation and the development of pubic hair in puberty. Therefore, it is not recommended to use FTSG from this area in permanently exposed areas like the face. Of note, in one of our study patients, an FTSG taken from the groin developed hyperpigmentation on one palm, while a second transplant taken from the opposite groin to the other palm did not exhibit this response.

In general, FTSG requires meticulous attention to detail and meticulous operative and postoperative care to achieve optimal outcomes [2,3]. However, the LOMA helps to facilitate this task, speeds up and safeguards the intraoperative process of full skin transplantation, and thus makes the process easier to apply.

Overall, the study’s results align with the experience of the physicians at the Department of Pediatric Surgery at the Klinikum Stuttgart, demonstrating the great functionality and applicability of the LOMA defatting machine leading to convincing results in skin quality and patient satisfaction.

It is important to acknowledge that the study was conducted at the department where the machine was developed, and the authors are employed there. The first author declares that there is no financial conflict of interest related to the LOMA. The LOMA machine’s patent is held by the head of the department, who is also one of the senior advisors of this publication. All patients evaluated for their FTSG were treated at the authors’ surgical department. These relationships are disclosed to ensure transparency in the research process.

## 5. Conclusions

Despite the availability of alternative treatment options, FTSGs remain highly valued for correcting scars on the face and hands, as well as for managing chronic and extensive wounds. However, the multitude of treatment options has led to a reduced focus on full-thickness skin grafting. This study underscores the enduring efficacy of full-thickness skin grafts, demonstrating their continued value in wound treatment and their ability to deliver excellent outcomes. This reaffirms their importance, particularly when applied appropriately. The LOMA system significantly aids in simplifying the procedural steps, thereby ensuring consistently high-quality results.

The comparison between control groups and transplanted patients provides valuable insights into the efficacy of full skin transplantation. While differences were noted in between-patient comparisons, the intra-patient comparison with the patient’s own healthy skin is more important, as it ensures that similar skin and body region characteristics are compared. Since no statistically significant differences were observed between healthy skin and LOMA-transplanted full skin, successful restoration of functionality in previously burnt areas was demonstrated convincingly.

Equally important, our study shows significant satisfaction among both patients and treating physicians regarding the outcomes of FTSG. Despite minor differences noted, the skin quality approaches that of healthy skin.

Full-thickness skin transplantation is a process that should be considered for patients requiring aesthetic and functional wound closure in high-stress and visible areas. Currently, FTSG may be underutilized due to its complex preparation process and the availability of other treatment options. However, this study shows that with the application of the innovative LOMA machine, the preparation process can be expedited, secured, and facilitated. This makes full-thickness skin grafting more accessible for a larger number of patients who require wound treatment with full recovery of skin functionality.

During the preparation of this work, the author used DeepL Write and ChatGPT 4o to correct language and spelling mistakes. After using these tools/services, the author(s) reviewed and edited the content as needed and take full responsibility for the content of the publication.

## Figures and Tables

**Figure 1 ebj-06-00044-f001:**
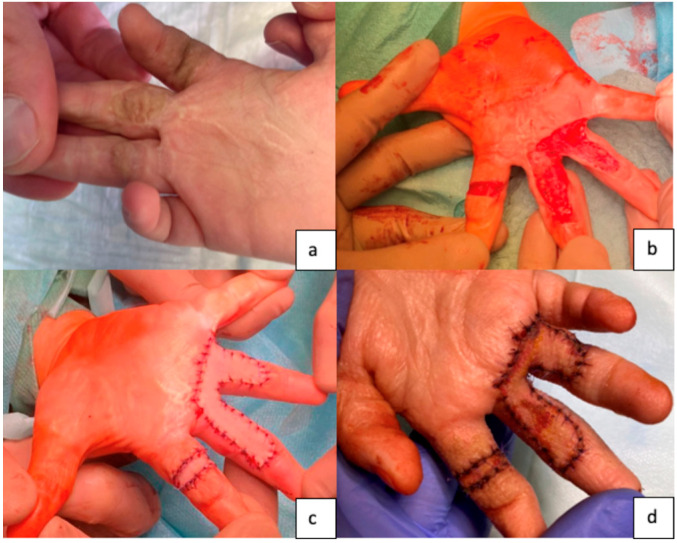
Preoperative and intraoperative images of a burned hand treated with full-thickness skin grafting performed abroad. (**a**) Preoperative condition of the burned hand; (**b**) intraoperative scar excision; (**c**) placement of the FTSG; (**d**) postoperative result at 2 weeks.

**Figure 2 ebj-06-00044-f002:**
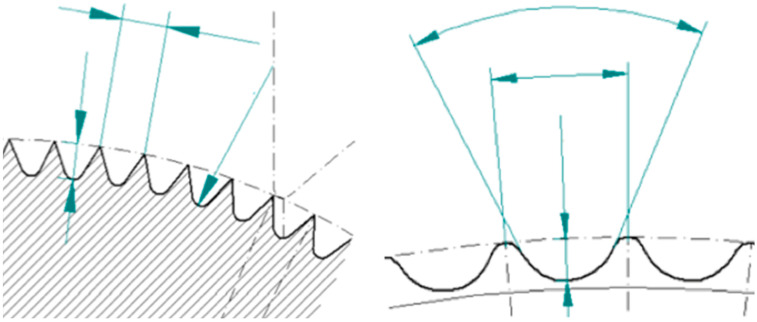
Sharp-edged crimper (**left**) and LOMA crimper with slightly rounded edges (**right**).

**Figure 3 ebj-06-00044-f003:**
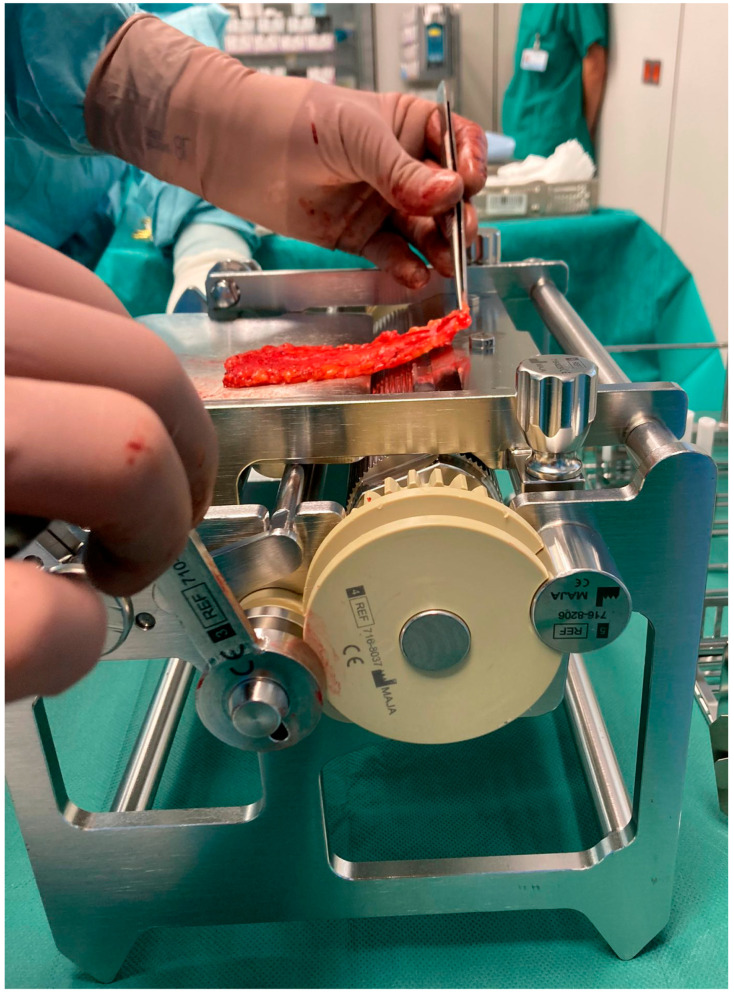
Application of the LOMA to intraoperatively defat a FTSG.

**Figure 4 ebj-06-00044-f004:**
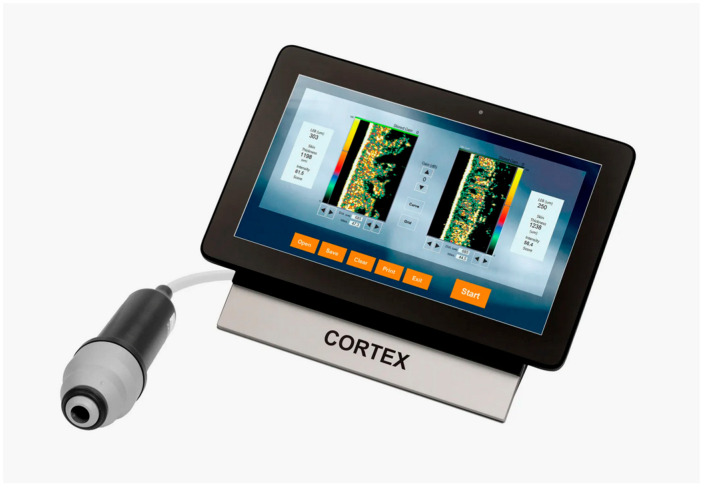
The DermaLab Combo with its high-resolution ultrasound probe [9].

**Figure 5 ebj-06-00044-f005:**
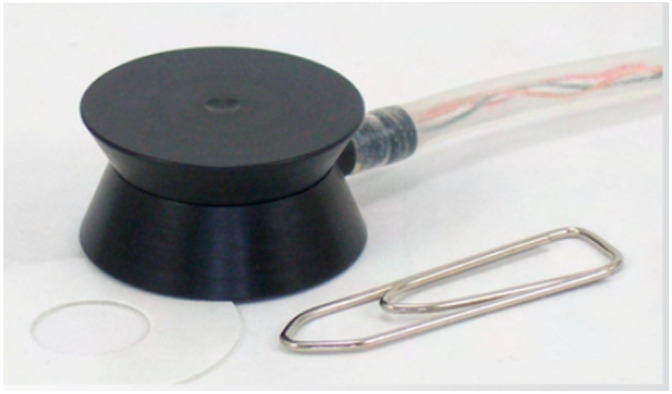
The DermaLab Combo skin elasticity probe [9].

**Figure 6 ebj-06-00044-f006:**
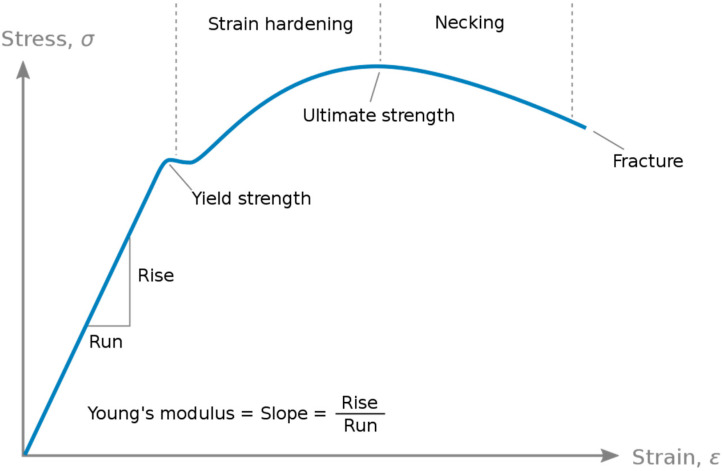
E represents the gradient of the linear segment on the stress–strain curve for a material subjected to tension or compression [10].

**Figure 7 ebj-06-00044-f007:**
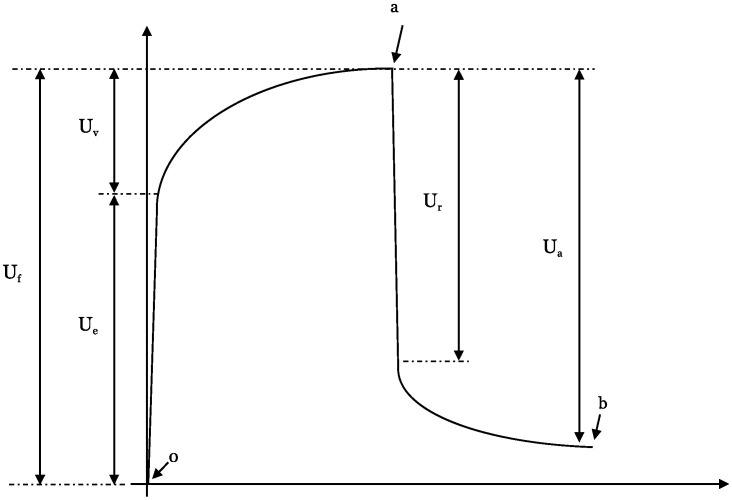
A single skin elasticity measurement curve subdivided into subsegments [7].

**Figure 8 ebj-06-00044-f008:**
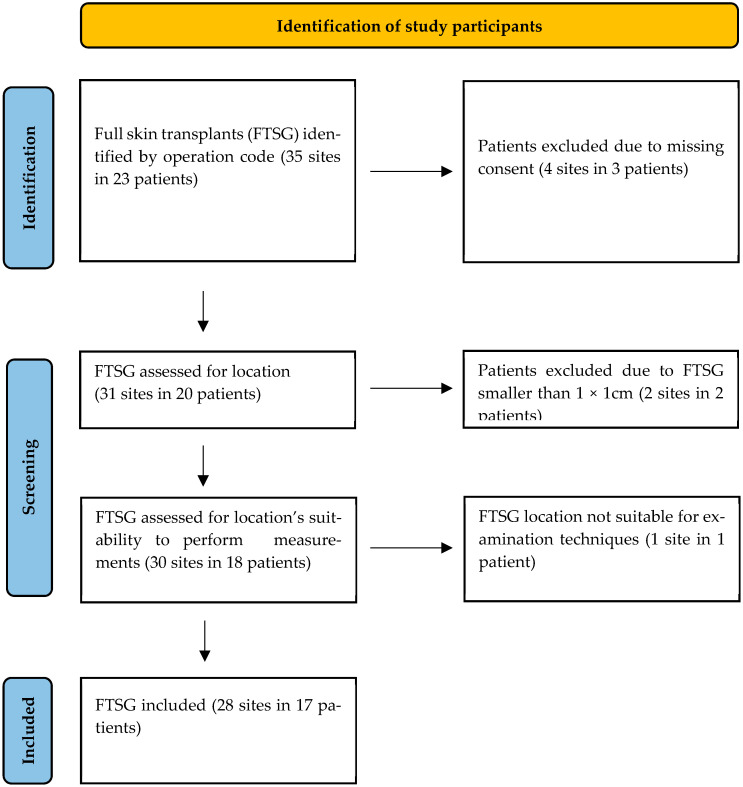
PRISMA flowchart for the identification of study participants.

**Figure 9 ebj-06-00044-f009:**
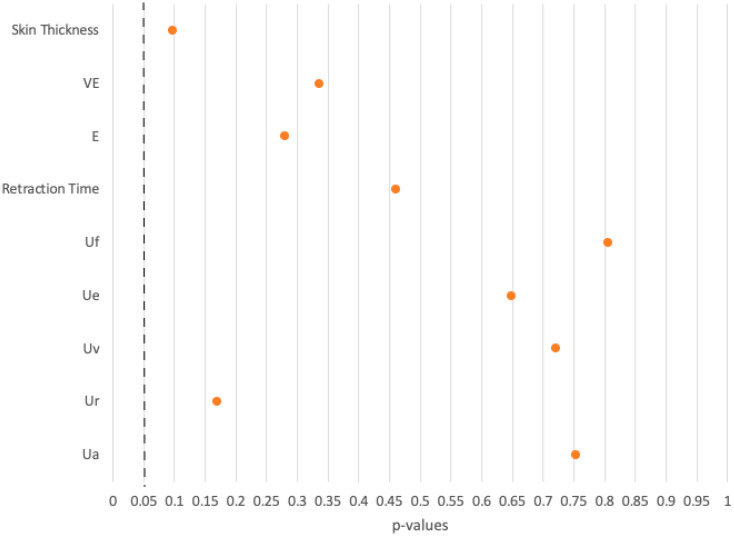
Overview of *p*-values of the comparison between the FTSG and the healthy opposite side groups.

**Figure 10 ebj-06-00044-f010:**
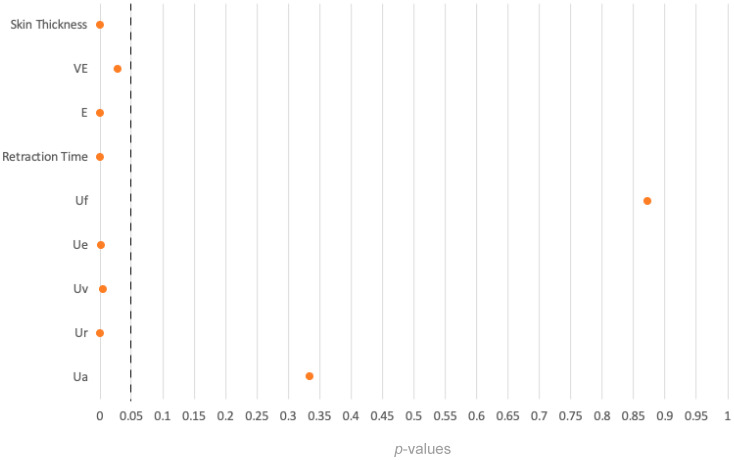
Overview of *p*-values of the comparison between the FTSG and the healthy palm groups.

**Figure 11 ebj-06-00044-f011:**
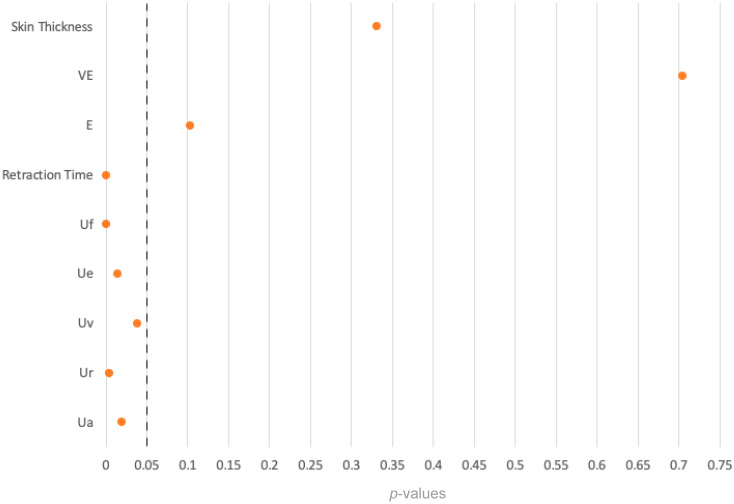
Overview of *p*-values of the comparison between the FTSG and the healthy ankle groups.

**Table 1 ebj-06-00044-t001:** Average POSAS scores for FTSG transplant patients based on patient questionnaires.

	Average POSAS	Median POSAS
Patient pain	1.9	1
Patient itch	1.5	1
Patient skin color	4.3	4
Patient skin stiffness	2.8	3
Patient skin thickness	2.9	3
Patient skin irregularity	2.9	2
Patient overall opinion	3.3	2.5

**Table 2 ebj-06-00044-t002:** Average POSAS scores for FTSG transplant based on observer questionnaires.

	Average POSAS	Median POSAS
Observer skin vascularity	1.1	1
Observer skin pigmentation	2.1	2
Observer skin thickness	2.2	2
Observer skin relief	2.0	2
Observer skin pliability	2.5	2
Observer skin surface area	1.9	2
Observer overall opinion	2.2	2

**Table 3 ebj-06-00044-t003:** Overview of the *p*-values for the measured U-values in comparison with the patients’ opposite side.

*U*-Values	*p*-Value
Uf	0.805
Ue	0.648
Uv	0.721
Ur	0.17
Ua	0.753

**Table 4 ebj-06-00044-t004:** Overview of the *p*-values for the measured U-values in the comparison with healthy patients’ palmar skin.

*U*-Values	*p*-Value
Uf	0.8728
Ue	0.0017 *
Uv	0.0054 *
Ur	0.0015 *
Ua	0.3347

* = significant result.

**Table 5 ebj-06-00044-t005:** Overview of *p*-values for the measured U-values in the comparison between the FTSG and the healthy ankle groups.

*U*-Values	*p*-Value
Uf	0.0005 *
Ue	0.0146 *
Uv	0.0377 *
Ur	0.0044 *
Ua	0.0198 *

* = significant result.

## Data Availability

The data that support the findings of this study are available from the corresponding author upon reasonable request. Due to ethical restrictions and privacy concerns, data are not publicly available.

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
