# Peer review of "Development of a Device for Defatting Full Skin Grafts Through Mechanical Defatting in Children and Adolescents"

_2673-1991, 2025, doi:10.3390/ebj6030044_

Round 1

Reviewer 1 Report

Comments and Suggestions for Authors

Hello Authors,

Thank you for your submission and novel device implementation.

A few "top level" updates that are needed:

1) Please review the formatting for the paper this includes the authorship list, conflicts of interest, and informed consent sections.

After reading the Author List and Affiliation guidelines it appears that the "Dr. med. univ" etc. would not be needed. 

2) Abstract is missing. See guidelines below.

  • Abstract: The abstract should be a total of about 200 words maximum. The abstract should be a single paragraph and should follow the style of structured abstracts, but without headings: 1) Background: Place the question addressed in a broad context and highlight the purpose of the study; 2) Methods: Describe briefly the main methods or treatments applied. Include any relevant preregistration numbers, and species and strains of any animals used; 3) Results: Summarize the article's main findings; and 4) Conclusion: Indicate the main conclusions or interpretations. The abstract should be an objective representation of the article: it must not contain results which are not presented and substantiated in the main text and should not exaggerate the main conclusions."

 Related to the article:

Introduction:

Lines 35-38 citation is needed

Line 41: The introduction appears to end without any formal transition into the proposed defatting device. I would strongly recommend a transitional piece before moving into the methods section.

Here are the Introduction guidelines Introduction: The introduction should briefly place the study in a broad context and highlight why it is important. It should define the purpose of the work and its significance, including specific hypotheses being tested. The current state of the research field should be reviewed carefully and key publications cited. Please highlight controversial and diverging hypotheses when necessary. Finally, briefly mention the main aim of the work and highlight the main conclusions. Keep the introduction comprehensible to scientists working outside the topic of the paper”

Methods:

Overall, was an ethics or IRB obtained for this study, individual study participant consent etc.?

Line 59: is LOMA the name of the device? Does LOMA stand for anything? Who/what is the source of LOMA. Is the LOMA an approved medical device from a regulatory body?

Lines 63-66 I would encourage putting this breakdown into a table.

Line 74 you state that both the palm and ankle were selected for measurements and as I read it the statement infers that the palm is only keratinized. Both locations are keratinized, additionally when you state constant pressure, what is meant by this? Lastly, would the ankle not moving by plantar or dorsiflexion infer some sort of tension being put on the skin?

Please note that you should note how and what statistical methods were utilized, including software.

Please provide details and an image of the DermaLab device with examples of the probes. 

Results:

Please include clinical examples of the data being presented, the images will assist in the varying results sections.

Discussion:

Lines 311-315 cite needed

344-345 I would add citations that reinforce this statement of skin property difference by location

A future note, in line 359 you mention some significant advantages to using LOMA. I might suggest a further economic impact to the medical system/clinic/hostile using this device with support this statement from a quantitative standpoint.

Lines 411-417: Please review the guidelines for Conflicts of Interest and the sections within the journal/article

References:

The references need to follow the journal format and requirements

The references list is very short for an original article submission; please add additional references from the field of study to substantiate the claims/statements and data within the article.

Reviewer 2 Report

Comments and Suggestions for Authors

  1. In one location, the authors refer to the Patient and Observer Scar Assessment Scale as POSAS, which is how I refer to it. However, more spots in the manuscript refer to it as the POTAS. This seems to be an error. 
  2. I had a hard time following the true goal of this paper. If the intent was to describe the effectiveness of the defatting machine at preparing full thickness skin grafts, the comparison group really should have been made up of full thickness skin grafts prepared the traditional way, by hand. Comparing skin quality of the transplanted skin to contralateral uninjured skin speaks more to full thickness skin grafts in general rather than those prepared by the author's preferred tool. 
  3. Why were full thickness skin grafts (presumably not all located on the palm) compared to healthy palmar and ankle skin other than the stated ease of access while clothed? It is well understood that skin on different areas of the body will demonstrate different properties and thus comparing grafted skin on a thigh, for example, with that on an ankle is difficult to use to draw any reliable conclusion.
  4. If one of the main goals of this paper is to describe the defatting tool used by the authors, images of this tool are needed. 

Round 2

Reviewer 1 Report

Comments and Suggestions for Authors

Dear authors thank you for the updates and enhancements for the paper. It is now ready for publication. 

Author Response

Thank you.